# Unlocking the Predictive Power of Nutritional Scores in Septic Patients

**DOI:** 10.3390/nu17030545

**Published:** 2025-01-31

**Authors:** Arianna Toscano, Federica Bellone, Noemi Maggio, Maria Cinquegrani, Francesca Spadaro, Francesca Maria Bueti, Giuseppe Lorello, Herbert Ryan Marini, Alberto Lo Gullo, Giorgio Basile, Giovanni Squadrito, Giuseppe Mandraffino, Carmela Morace

**Affiliations:** 1Unit of Internal Medicine, Department of Clinical and Experimental Medicine, University Hospital G. Martino, University of Messina, 98100 Messina, Italy; arianna.toscano13@gmail.com (A.T.); fbellone@unime.it (F.B.); noemimaggio05@gmail.com (N.M.); mariacinquegrani@gmail.com (M.C.); francescaspadaro@yahoo.com (F.S.); francescabueti085@gmail.com (F.M.B.); giuseppelorello@alice.it (G.L.); hrmarini@unime.it (H.R.M.); gsquadrito@unime.it (G.S.); carmela.morace@unime.it (C.M.); 2Unit of Rheumatology, Azienda Ospedaliera Papardo, 98158 Messina, Italy; albertologullo@virgilio.it; 3Unit of Geriatrics, Department of Biomedical and Dental Science and Morphofunctional Imaging, University Hospital G. Martino, University of Messina, 98100 Messina, Italy; basileg@unime.it

**Keywords:** sepsis, septic shock, nutritional status, nutritional scoring, adverse outcome, modified Glasgow Prognostic Score (mGPS), Prognostic Nutritional Index (PNI), NUTRIC, CONUT, BUN-to-albumin ratio (BAR), internal medicine unit

## Abstract

**Background**: Sepsis is a critical condition characterized by severe immune dysregulation, ranking among the leading causes of morbidity and mortality in intensive care and internal medicine units. Nutritional status plays a pivotal role in modulating these responses, as when inadequate it can compromise immune defenses, the body’s ability to handle stress and inflammation, and the clinical course. Malnutrition is frequently observed in septic patients and is strongly associated with worse clinical outcomes, including increased mortality, prolonged hospital stays, and greater complication rates. In this context, nutritional scoring systems have emerged as valuable tools to evaluate patients’ nutritional status and predict clinical trajectories. **Objectives**: Given the absence of a direct comparison of their performance in an internal medicine setting, this study aimed to assess the effectiveness of various nutritional scores as predictive tools for clinical outcomes in septic patients, emphasizing their application within the field of internal medicine. **Methods and Results**: A retrospective analysis was conducted on 143 patients diagnosed with sepsis or septic shock who were admitted to an internal medicine unit. Key variables included clinical and laboratory parameters, comorbidities, and nutritional scores at the time of diagnosis. The modified Glasgow Prognostic Score (mGPS), the Prognostic Nutritional Index (PNI), the Controlling Nutritional Status (CONUT) score, the modified Nutrition Risk in Critically Ill (mNUTRIC) score, and the blood urea nitrogen-to-albumin ratio (BAR) were evaluated in forecasting mortality and clinical outcomes in patients with sepsis. Among them, the mNUTRIC score emerged as the strongest independent predictor of in-hospital mortality, with a good performance and a reasonable threshold for risk stratification. **Conclusions**: The study highlights the mNUTRIC score’s practicality and reliability in assessing nutritional and inflammatory risks in septic patients, particularly in non-ICU settings. These findings suggest its potential utility in guiding nutritional interventions and improving clinical outcomes, emphasizing the importance of integrating nutritional assessment into sepsis management.

## 1. Introduction

Sepsis is a heterogeneous, complex, and life-threatening clinical condition characterized by an intense and dysregulated systemic inflammatory response to the presence of pathogenic microorganisms [1,2].

Despite significant therapeutic advances, sepsis and septic shock remain major global health issues, affecting millions annually and resulting in exceptionally high death rates [3,4,5]. Unfavorable outcomes are reported in one in three to one in six individuals diagnosed with sepsis [3,4,5].

The evaluation of nutritional status has been widely recognized in the literature as a predictor of negative outcomes and longer hospital stays, as it reflects not only the patient’s baseline health but also their ability to withstand and recover from critical illnesses. Commonly used indicators, such as body mass index (BMI) and serum albumin concentration, while practical, present several limitations. For example, body weight and body composition metrics, and circumferences, can be falsely elevated in the presence of conditions like peripheral edema, body cavity effusions, and anasarca. Similarly, serum albumin levels may be influenced by factors beyond nutritional status, such as hepatic synthesis and catabolism dynamics [6,7,8]. Therefore, these indicators, while valuable, require integration with other parameters to enhance their reliability and provide a more comprehensive assessment of the patient’s prognostic outlook.

Over the past decades, several scores have been developed to evaluate nutritional status across different patient populations. Among the most clinically relevant and easily applicable in practice are the modified Glasgow Prognostic Score (mGPS), the Prognostic Nutritional Index (PNI), the Controlling Nutritional Status (CONUT) score, and the modified Nutrition Risk in Critically Ill (NUTRIC) score.

The modified Glasgow Prognostic Score (mGPS), which combines serum albumin and C-reactive protein (CRP) levels to evaluate the interplay between nutrition and systemic inflammation, has gained attention for its simplicity and prognostic value and has been extensively validated in oncology and other chronic inflammatory conditions, showing robust correlations with survival outcomes [9,10].

The Prognostic Nutritional Index (PNI), initially developed to evaluate the nutritional status of patients undergoing surgery, combines serum albumin concentration and total lymphocyte count to offer a straightforward measure of nutritional and immune status which can play a role in patients’ outcomes [11,12,13,14,15,16,17,18]. The Controlling Nutritional Status (CONUT) score integrates serum albumin, total lymphocyte count, and total cholesterol levels, allowing for a nuanced assessment of malnutrition severity [19]. The CONUT score has demonstrated prognostic utility across several clinical scenarios, including cancer [20], coronary artery disease [21], atrial fibrillation [22], acute heart failure [23], and stroke [24]. Recently, Miano et al. highlighted the CONUT score’s high prognostic value for in-hospital mortality, risk of sepsis, and longer hospital stays in internal medicine patients, recommending its use as a nutritional screening tool to identify high-risk individuals [25].

The Nutrition Risk in Critically Ill (NUTRIC) score—introduced by Heyland et al. in 2011—integrates age, number of comorbidities, days from hospital to ICU admission, SOFA (Sequential Organ Failure Assessment), APACHE II (Acute Physiology and Chronic Health Evaluation), and interleukin-6 level (IL-6) as an optional variable and is the first validated nutritional score for critically ill patients [26]. To address the practical challenge of measuring IL-6, a simplified version, the modified NUTRIC score (mNUTRIC), was later developed [27]. Subsequent studies have consistently confirmed the NUTRIC and mNUTRIC scores’ roles as predictors of adverse outcomes across various clinical settings [28,29,30,31,32,33,34,35]; moreover, all but one study conducted in critically ill cirrhotic patients, in which NUTRIC had a moderate prognostic advantage in comparison to mNUTRIC [36], did not find the NUTRIC score to be superior to the mNUTRIC score in terms of mortality prediction [28,29,30,31].

Finally, a novel, non-invasive, and accessible prognostic tool based on the blood urea nitrogen (BUN)-to-serum albumin ratio (BAR) has been proposed. This ratio has shown promise in predicting outcomes in sepsis, as it combines markers of kidney function and protein status to provide a simple yet effective measure of systemic health. Studies suggest that higher BAR values are associated with increased mortality and poorer clinical outcomes, making it a valuable addition to the array of nutritional and prognostic tools in sepsis management. Moreover, its ease of calculation and reliance on routine laboratory parameters make it particularly suitable for widespread clinical use [8,37,38,39,40,41,42].

In recent years, significant attention has been devoted to understanding the role of malnutrition in patients admitted to intensive care units (ICUs). According to the European Society for Clinical Nutrition and Metabolism (ESPEN) guidelines, critically ill patients who remain in the ICU for more than 48 h should be considered at risk for malnutrition, irrespective of predisposing risk factors [43]. This underscores the high prevalence of malnutrition among ICU patients and its potential impact on clinical outcomes.

A study by Bodolea et al. investigated the predictive value of several nutritional scores, including the PNI, CONUT, NUTRIC, and its modified version (mNUTRIC), in assessing the need for mechanical ventilation and in-hospital mortality among patients with severe COVID-19 and acute respiratory distress syndrome (ARDS) admitted to the ICU [44]. The findings revealed that these non-invasive biological scores were effective independent predictors of in-hospital mortality in this population, highlighting their utility in clinical practice; a direct comparison between those scores was, however, not performed in this study [44].

Subsequently, Zou et al. conducted a study aimed at evaluating the efficacy of six nutritional scoring systems, including Nutritional Risk Screening 2002 (NRS-2002), NUTRIC, modified NUTRIC, PNI, CONUT, and the TCB index (TCBI), in predicting prognosis in COVID-19 patients [45]. Their results demonstrated that the mNUTRIC score and NRS-2002 were the most effective prognostic tools for all COVID-19 patients and ICU-admitted COVID-19 patients, respectively [45].

Additionally, the relationship between nutritional status and clinical outcomes in ICU patients was assessed in another study by Moghaddam et al., who utilized various nutritional risk assessment tools to identify the most accurate predictive model [46]. The mNUTRIC score outperformed the other tools in predicting clinical outcomes, likely due to its specificity in a critical care setting [46].

Despite these advancements, no direct comparison has yet been performed between different nutritional scores to assess their predictive value in septic patients admitted to internal medicine wards, leaving a critical gap in our understanding of their utility in this specific setting.

This study sought to systematically assess the utility of these scores in forecasting key clinical outcomes, including mortality, in septic patients admitted to an internal medicine ward. Moreover, the study further aimed to determine specific cutoff values for the scores predicting the clinical outcome, if any, enabling improved predictive accuracy and the establishment of practical thresholds for risk stratification in this patient population. The findings may offer clinicians valuable tools to tailor nutritional interventions and potentially enhance patient outcomes.

## 2. Materials and Methods

### 2.1. Patients’ Selection and Data Collection

The medical records of 1123 patients consecutively admitted to the Internal Medicine Unit of the “Gaetano Martino” University Hospital in Messina between 1 January 2020 and July 2024 were reviewed. Records with “Sepsis” or “Septic Shock” listed as admission or discharge diagnoses were further examined to identify patients meeting the Third International Consensus Definitions for Sepsis and Septic Shock (Sepsis-3). The final study population included 143 patients.

The following variables were collected for these patients: age at admission, length of hospital stay, and mortality outcomes. Vital signs and consciousness level were documented at the time of sepsis diagnosis, including blood pressure, mean arterial pressure (MAP), heart rate, respiratory rate, and body temperature. Data from arterial blood gas analysis, oxygen therapy regimen (if applicable), and lactate levels were also recorded. Information regarding the source of sepsis, culture tests performed, and outcomes, including isolated pathogens, were also collected.

Finally, data on comorbidities were gathered, including the presence of arterial hypertension, diabetes mellitus, chronic kidney disease (with staging), heart failure (classified according to the New York Heart Association [NYHA]), liver cirrhosis, and chronic obstructive pulmonary disease (COPD).

Laboratory data at the time of sepsis or septic shock diagnosis were recorded, including complete blood count parameters, blood urea nitrogen (BUN, calculated as 47% of urea nitrogen using the formula: serum urea/2.1428), creatinine, electrolytes (sodium, potassium, chloride, and calcium), albumin, total and direct bilirubin, international normalized ratio (INR), and lipid profile (total cholesterol, HDL cholesterol, triglycerides, and LDL cholesterol). The estimated glomerular filtration rate (eGFR) was calculated using the Chronic Kidney Disease Epidemiology Collaboration (CKD-EPI) 2021 formula.

### 2.2. Nutritional Scores

For each patient, the neutrophil-to-lymphocyte ratio (NLR) and platelet-to-lymphocyte ratio (PLR) were calculated. Nutritional and prognostic scores, including the modified Glasgow Prognostic Score (mGPS), Prognostic Nutritional Index (PNI), modified NUTRIC (mNUTRIC) score, CONUT score, and blood urea nitrogen-to-albumin ratio (BAR), were assessed at the time of sepsis diagnosis.

The mGPS was determined by evaluating serum C-reactive protein (CRP) levels as combined with albumin concentrations. A score of 0 was assigned to patients with CRP levels ≤ 1 mg/dL, regardless of their albumin concentration. Patients with CRP levels > 1 mg/dL were scored based on their albumin concentration: those with albumin ≥ 3.5 g/dL received a score of 1, while those with albumin < 3.5 g/dL were assigned a score of 2. Hence, higher mGPS values reflected a combination of elevated inflammation and decreased nutritional reserves.

The PNI was calculated as 10 × serum albumin concentration (g/dL) + 0.005 × lymphocyte count (mm^3^). A higher PNI was considered as an indicator of better nutritional and immunological status, while a lower PNI was considered to suggest malnutrition and potential immune compromise.

The CONUT score was calculated and used to quantify malnutrition severity as follows (Table 1):

The mNUTRIC score was calculated using a validated online calculator available at MDCalc [https://www.mdcalc.com/calc/4011/nutrition-risk-critically-ill-nutric-score, last access on 20 August 2024]. The same calculator was used for all patients to ensure consistency in the scoring process.

The BAR was calculated by dividing the blood urea nitrogen (BUN) concentration, expressed in mg/dL, by the serum albumin concentration, expressed in g/dL.

### 2.3. Statistical Analysis

The statistical analysis was performed using SPSS (Statistical Package for the Social Sciences) software, version 26 (IBM SPSS version 26.0 64 bit, IBM corp 2019). Normality was tested by the Kolmogorov–Smirnoff test. Since several variables showed a non-normal distribution, a classic non-parametric approach was used. Continuous variables were expressed as medians and interquartile ranges (IQRs) and categorical variables as percentages. Comparative analyses were performed using appropriate non-parametric tests (the Mann–Whitney test for continuous variables and the chi-squared test for categorical variables). A multivariable stepwise regression analysis was designed in order to verify if any nutritional score—among those considered and estimated—was able to predict the adverse outcome in septic patients admitted to the internal medicine ward. Moreover, since one score only showed a significant predictive ability, an ROC analysis was also performed to detect a predictive threshold in the studied setting. Comorbidities were also considered in a logistic regression model to evaluate the impact of each considered disease on mortality, and the number of comorbidities was considered in a simple regression model. According to the cross-sectional design of the study, no comparisons or repeated measures were scheduled or performed; thus, no power analysis was needed. We also scheduled to compare the nutritional status as evaluated by the scores according to the presence of any comorbidity significantly associated with exitus to verify if any score is able to detect the difference in nutritional status in this type of patient. A level of *p* < 0.05 was chosen to denote the statistical significance where appropriate.

## 3. Results

### 3.1. Study Population Characteristics

The main characteristics of the 143 patients (43.4% women) are described below and summarized in Table 2. The median age at admission was 79.3 years (IQR: 12.94), and the median length of hospital stay was 12 days (IQR: 12).

At admission, vital signs were as follows: median systolic blood pressure: 110 mmHg (IQR: 40); median diastolic blood pressure: 60 (IQR: 20); median heart rate: 90 bpm (IQR: 31); median respiratory rate: 20 breaths per minute (IQR: 7); and median temperature: 37.2 °C (IQR: 0.8). The median Glasgow Coma Scale (GCS) score was 14 (IQR: 2).

### 3.2. Comorbidities and Sepsis-Related Parameters

Comorbidities were highly prevalent in this cohort. The most common conditions included hypertension (AH; 99/143, 69.2%), diabetes mellitus (T2DM; 62/143, 43.4%), history of coronary cardiovascular events (CAD; 47/143, 32.9%), and chronic heart failure (HF; 56/143, 39.2%). Additionally, 36 patients (25.2%) had chronic obstructive pulmonary disease (COPD), while 81 patients (56.6%) were suffering from chronic kidney disease (CKD), including 9 with end-stage renal disease (ESRD; 6.3%). Liver cirrhosis was documented in 21 patients (14.7%). Most patients (122 out of 143, i.e., 85.3%) presented with more than one associated comorbidity.

The logistic regression model suggested CKD (including ESRD) as the only comorbidity associated with the adverse outcome compared with AH, T2DM, CAD, and HF (OR: 3.139, CI: 1.565–6.295; *p* < 0.001).

The median PCT was 3.4 ng/mL (IQR: 11.57). The median hsCRP was 12.65 mg/dL (IQR: 13.06). The PLR score had a median of 158.48 (IQR: 151.02). The APACHE II score at the time of sepsis diagnosis recorded a median of 20 (IQR: 8), while the SOFA score at the time of sepsis diagnosis had a median of 5 (IQR: 4).

The primary focus of infection was identified as the lungs in 35 cases (24.5%), followed by urinary tract infections in 32 cases (22.4%). In 49 patients (34.3%), it was not possible to isolate a specific pathogen. Among those with identified pathogens, a single pathogen was isolated in 53 cases (37.1%), two pathogens in 29 cases (20.3%), and three pathogens in 3 cases (2.1%). Among the pathogens isolated, *Escherichia coli* (*E. coli*) was the most frequently identified, accounting for 24 cases. This was followed by *Acinetobacter baumannii* (13 cases), *Klebsiella pneumoniae* (11 cases), *Proteus mirabilis* (8 cases), and *Staphylococcus aureus* (8 cases). Other pathogens included *Pseudomonas aeruginosa* (five cases), *Staphylococcus haemolyticus* (three cases), *Enterococcus faecalis* (three cases), and *Staphylococcus epidermidis* (two cases). Rare isolates included *Moraxella catarrhalis*, *Streptococcus pneumoniae*, *Bacteroides*, and SARS-CoV-2, each detected in one case. The isolated pathogens are represented in Table 3.

In-hospital mortality was notably high, with 70 patients (49.0%) dying during their hospital stay; however, this aligned with the literature data.

### 3.3. Nutritional and Severity Scores

Concerning the nutritional and severity scores, the mGPS had a median of 2 (IQR: 0), the PNI score was 33.92 (IQR: 9.65), the CONUT score had a median of 8 (IQR: 3), the mNUTRIC score recorded a median of 4 (IQR: 3), and the BAR score had a median of 15.07 (IQR: 13.24).

At the time of sepsis diagnosis, the following variables were analyzed: mGPS, PNI, CONUT, mNUTRIC, and BAR. A stepwise multivariable regression analysis was performed to evaluate the association of these variables with in-hospital mortality. Among the variables tested, the mNUTRIC score emerged as a significant independent predictor of mortality.

To further assess its predictive performance, an ROC curve analysis was conducted, yielding an area under the curve (AUC) of 0.814 (95% CI: 0.737–0.891), indicating good discriminatory ability. The analysis identified a cut-off value of 4.5 points for the mNUTRIC score as the optimal threshold for predicting mortality, based on the Youden’s index, which was calculated at 0.508 (Figure 1).

## 4. Discussion

This study evaluated the utility of various nutritional scores in predicting key clinical outcomes, including mortality, in septic patients admitted to an internal medicine ward. Among the variables analyzed, the mNUTRIC score emerged as the strongest predictor of in-hospital mortality, with an AUC of 0.814, underscoring its ability to stratify risk effectively in septic patients. An additional analysis was performed to establish specific cutoff values for the mNUTRIC score in patients admitted to internal medicine wards, aimed at enhancing its predictive accuracy. A threshold of 4.5 points was identified as the optimal cutoff, thereby providing a clear criterion for clinical application.

In this cohort, the high in-hospital mortality rate of 49.0% reflects the severity of sepsis and is consistent with the expected outcomes for septic patients, particularly when considering the high prevalence of comorbidities. The pathogen distribution observed in this study further underscores the complexity of sepsis management. Gram-negative bacteria, particularly *Escherichia coli* and *Acinetobacter baumannii*, were the most frequently isolated pathogens, consistent with established patterns in sepsis and previous research on nosocomial infections [5]. However, the inability to isolate specific pathogens in 34.3% of cases highlights the diagnostic challenges in sepsis care and reinforces the importance of empirical antimicrobial regimens tailored to local epidemiological trends [47].

Over 85% of our cohort’s patients had more than one underlying condition, with hypertension, diabetes mellitus, chronic kidney disease, and chronic heart failure being the most common. These findings align with prior studies that highlight the cumulative impact of multimorbidity on sepsis outcomes [2,5]. Notably, in our cohort, logistic regression analysis identified CKD, including end-stage renal disease (ESRD), as the only comorbidity significantly associated with adverse outcomes, with an odds ratio of 3.139 (95% CI: 1.565–6.295; *p* <0.001). This aligns with the existing literature findings demonstrating that CKD is independently associated with increased mortality in both septic and non-septic populations [47]. Beyond its known contributions to systemic inflammation, oxidative stress, and cardiovascular complications, which elevate mortality risk [47,48,49], CKD also creates a complex pathophysiological environment marked by metabolic alterations that affect nutrient intake, metabolism, and energy expenditure [50]. These changes predispose patients to malnutrition, further exacerbating inflammation and increasing the risk of adverse outcomes.

Effective management of septic patients needs an integrated approach that addresses both acute and chronic disease processes while considering the local pathogen distribution. Malnutrition, often overlooked in this context, is another critical factor that could significantly affect outcomes in patients hospitalized with sepsis, particularly in those with multimorbidity. The nutritional scores evaluated in this study provided complementary perspectives on patient risk. The mGPS, the PNI, and CONUT incorporate markers of inflammation and malnutrition. Similarly, the BAR score, calculated using blood urea nitrogen and albumin, has shown promise in predicting mortality in septic patients [38,42] as well as in other settings [8,37,39,40,41].

However, the results of our study did not identify any of these scores as consistently useful in this specific patient population. These findings stand in contrast to previous research that has reported significant predictive value for these tools in other settings, particularly among critically ill patients. This discrepancy highlights the complexity of sepsis and its multifaceted interaction with underlying comorbidities, organ dysfunction, and nutritional status. It also underscores the challenges of translating predictive models validated in ICU populations to more heterogeneous cohorts, such as those in internal medicine wards.

The mNUTRIC score, indeed, was found to be associated with the outcome in our study; thus, this score emerges as a practical tool for assessing nutritional risk in this population, as it considers the interplay between malnutrition and illness severity. Originally developed and validated for use in critically ill patients in the ICU setting, the mNUTRIC score has since been evaluated in diverse clinical settings. Its initial validation demonstrated its ability to stratify ICU patients at high nutritional risk who would benefit most from targeted nutritional interventions [26]. Moreover, it has consistently outperformed other nutritional scoring systems, as evidenced by studies conducted on critically ill populations [45,46]. Additional research has reinforced the mNUTRIC score’s ability to predict adverse outcomes in high-risk populations [27,28,29,33]. These findings support the notion that the mNUTRIC score can be a valuable tool beyond the ICU, addressing nutritional risk and guiding interventions in several healthcare settings. Its simplicity, based on routinely available clinical parameters, makes it particularly advantageous for use outside the ICU, where resource limitations may preclude more complex assessments. Furthermore, the mNUTRIC score’s focus on both malnutrition and inflammation underscores its relevance in patients with conditions such as sepsis and multimorbidity, where systemic inflammation plays a pivotal role in nutritional decline and overall prognosis. Our findings support its potential applicability in internal medicine wards, where patients often present with significant comorbidities and disease complexity. This broader use of the mNUTRIC score could facilitate early risk identification and intervention in a wider range of healthcare settings, extending its clinical impact beyond the ICU.

Despite its strengths, including being the first study in the literature to perform a comprehensive evaluation of various nutritional and inflammatory scoring systems in septic patients admitted to an internal medicine ward, this study has several limitations. The retrospective, single-center design restricts the generalizability of the findings, while the cross-sectional analysis fails to account for dynamic changes in nutritional and inflammatory scores over time. The reliance on readily available clinical and laboratory data also precluded the inclusion of more nuanced markers of disease progression, which could have provided additional insight into the interplay between nutrition, inflammation, and outcomes in sepsis. Finally, in hospitalized patients, variables such as body mass index (BMI) and body circumferences may be less reliable or even misleading due to factors like fluid retention, acute inflammation, or rapid shifts in body composition that obscure true nutritional status. Nonetheless, the absence of these measures in our analysis represents a limitation, as they could still contribute to a more comprehensive understanding of the relationship between nutritional status and outcomes.

Future research should focus on prospective, multicenter studies to validate these findings and explore the impact of interventions guided by nutritional scores on clinical outcomes. Incorporating comprehensive anthropometric and biochemical measures, including BMI and other markers of nutritional status, could further enhance the accuracy and applicability of these tools in septic populations.

## 5. Conclusions

In conclusion, this study highlights the significant prognostic utility of nutritional status evaluation in septic patients admitted to internal medicine wards. Among the tools evaluated, the mNUTRIC score—likely because of its dual focus on malnutrition and illness severity—emerged as the most robust predictor of in-hospital mortality, demonstrating its capacity to effectively stratify risk and guide clinical decision-making. The identification of a specific cutoff value of 4.5 points for the mNUTRIC score enhances its clinical applicability, providing a clear threshold for assessing nutritional risk in septic patients. Its simplicity and reliance on readily available clinical parameters make it accessible in resource-limited settings and suitable for early identification of high-risk patients, potentially guiding timely nutritional and therapeutic interventions. Importantly, our study supports the expansion of its utility beyond the ICU, highlighting its applicability in internal medicine settings.

While this study provides important insights, its retrospective, single-center design and the exclusion of dynamic changes in nutritional and inflammatory markers over time are limitations. Future research should focus on prospective, multicenter studies to validate these findings and explore the impact of interventions guided by nutritional scores on clinical outcomes.

## Figures and Tables

**Figure 1 nutrients-17-00545-f001:**
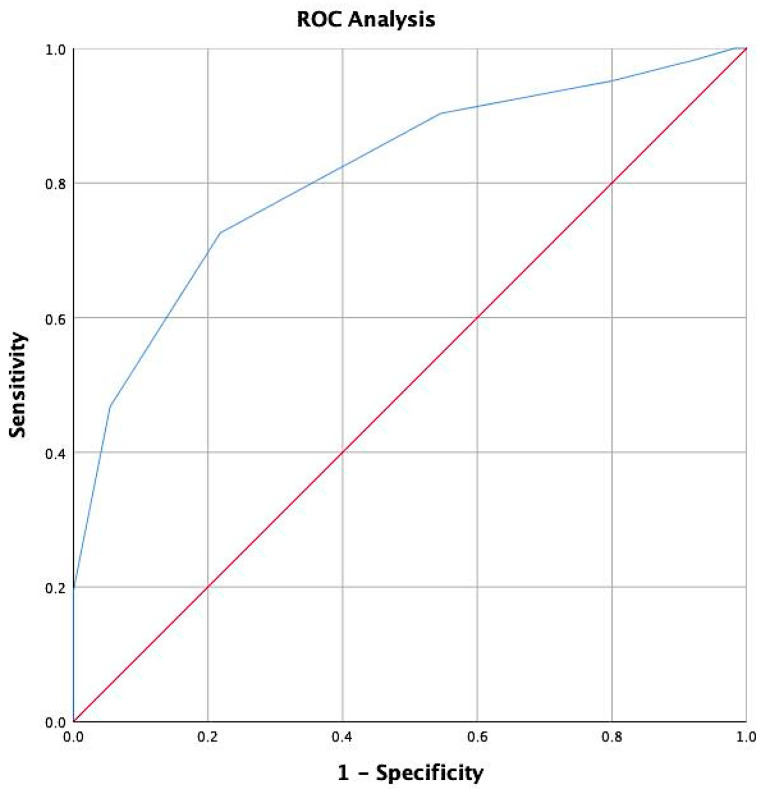
ROC curve illustrating the performance of the model testing mNUTRIC on exitus.

**Table 1 nutrients-17-00545-t001:** CONUT score, adapted from Ref. [19].

Variable	Normal	Light	Moderate	Severe
Serum albumin (g/dL)	3.5–4.5	3.0–3.49	2.5–2.9	<2.5
Albumin score	0	2	4	6
Total lymphocyte count (mm^3^)	≥1600	1200–1599	800–1199	<800
Total lymphocyte count score	0	1	2	3
Total cholesterol (mg/dL)	>180	140–180	100–139	<100
Total cholesterol score	0	1	2	3
CONUT score	0–1	2–4	5–8	9–12
Assessment	Normal	Light	Moderate	Severe

**Table 2 nutrients-17-00545-t002:** Characteristics of the study population.

	Total: 143 Patients
**Patient characteristics**	
Age, years	79.3 (12.94)
Women, n [%]	62 [43.4%]
Length of stay (days)	12 (12)
**Vital signs**	
SBP (mmHg)	110 (40)
DBP (mmHg)	60 (20)
MAP (mmHg)	76.7 (23.3)
HR (bpm)	90 (31)
RR (bpm)	20 (7)
BT (°C)	37.2 (0.8)
GCS (points)	14 (2)
**Laboratory data**	
** *Hematological profile* **	
Hb (g/dL)	10.4 (3.4)
HCT (%)	31 (10.3)
WBC (cells/mmc)	13,950 (12,538)
N (%)	84 (8)
L (%)	10 (7)
N/L	8.4 (6.8)
Platelets (cells/mmc)	196,000 (159,000)
PLR	158.5 (151)
** *Renal function markers* **	
Urea (mg/dL)	87 (37.3)
BUN (mg/dL)	41.3 (37.3)
Serum creatinine (mg/dL)	1.7 (2)
eGFR (CKD EPI 2021, mL/min)	34.9 (41.5)
Uric acid (mg/dL)	7.2 (4.4)
Sodium (mmol/L)	137 (8)
Potassium (mmol/L)	4 (1.3)
Calcium (mg/dL)	8.2 (0.9)
Chloride (mmol/L)	101 (7)
** *Liver function tests (LFTs)* **	
AST (U/L)	27 (42)
ALT (U/L)	20 (31)
INR	1.27 (0.34)
Serum albumin (g/dL)	2.67 (0.65)
** *Lipid panel* **	
Total cholesterol (mg/dL)	110 (51)
HDL cholesterol (mg/dL)	27.5 (22)
Triglyceride (mg/dL)	118 (80)
LDL cholesterol (mg/dL)	54.4 (43.4)
NT-proBNP (ng/mL)	6517 (15,141)
Plasma glucose (mg/dL)	134.5 (129)
**Inflammatory and sepsis-related indicators**	
hs-CRP (mg/dL)	12.6 (13)
PCT (ng/mL)	3.4 (11.6)
Lactate (mmol/L)	2 (1.4)
APACHE	20 (8)
SOFA	5 (4)
qSOFA	1 (1)
**Nutritional scores**	
mGPS	2 (0)
PNI	33.9 (9.6)
CONUT	8 (3)
mNUTRIC	4 (3)
BAR	15.1 (13.2)

Data are presented as medians (interquartile ranges) or numbers and percentages [%]. Abbreviations: SBP: systolic blood pressure; DBP: diastolic blood pressure; MAP: mean arterial pressure; HR: heart rate; RR: respiratory rate; BT: Body Temperature; GCS: Glasgow Coma Scale; BUN: blood urea nitrogen; eGFR: estimated glomerular filtration rate; PCT: procalcitonin; hs-CRP: high-sensitivity C-reactive protein; Hb: hemoglobin; HCT: hematocrit; WBC: white blood cell; N: neutrophil, L: lymphocyte; N/L: neutrophil/lymphocyte ratio; NT-pro-BNP: N-terminal pro-B-type natriuretic peptide; AST: aspartate aminotransferase; ALT: alanine aminotransferase; INR: International Normalized Ratio; APACHE: Acute Physiology and Chronic Health Evaluation; qSOFA: quick Sequential Organ Failure Assessment; SOFA: Sequential Organ Failure Assessment; mGPS: modified Glasgow Prognostic Score; PNI: Prognostic Nutritional Index; CONUT: Controlling Nutritional Status; mNUTRIC: modified Nutrition Risk in the Critically Ill score; BAR: BUN-to-albumin ratio; PLR: platelet-to-lymphocyte ratio. Round brackets used for IQRs; square brackets used for percentages.

**Table 3 nutrients-17-00545-t003:** Isolated pathogens and their frequencies.

Pathogen	Cases	Percentage (%)
*Escherichia coli*	24	27.0%
*Acinetobacter baumannii*	13	14.6%
*Klebsiella pneumoniae*	11	12%
*Proteus mirabilis*	8	9.0%
*Staphylococcus aureus*	8	9.0%
*Pseudomonas aeruginosa*	5	5.6%
*Streptococcus haemolyticus*	3	3.4%
*Enterobacter faecalis*	3	3.4%
*Streptococcus epidermidis*	2	2.2%
*Enterobacter cloacae*	1	1.1%
*Staphylococcus hominis*	1	1.1%
*Streptococcus sanguinis*	1	1.1%
*Moraxella catarrhalis*	1	1.1%
*Streptococcus pneumoniae*	1	1.1%
*Staphylococcus ludgunensis*	1	1.1%
*Raoultella ornithinolytica*	1	1.1%
*Enterococcus faecium*	1	1.1%
*Staphylococcus* spp.	1	1.1%
*Bacteroides* spp.	1	1.1%
*Enterobacter* spp.	1	1.1%
SARS-CoV-2	1	1.1%

## Data Availability

Dataset available on request from the authors. The raw data supporting the conclusions of this article will be made available by the authors on request.

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
