# Peer review of "Unlocking the Predictive Power of Nutritional Scores in Septic Patients"

_nutrients, 2025, doi:10.3390/nu17030545_

Round 1
Reviewer 1 Report
Comments and Suggestions for Authors
This is an important study.
Introduction
Please present your literature search so researchers who build on your research can update it.
You described the indices you planned to compare:
“Over the past decades, several scores have been developed to evaluate the nutritional status across different patients’ populations. Among the most clinically relevant and easily applicable in practice are the modified Glasgow Prognostic Score (mGPS), the Prognostic Nutritional Index (PNI), the Controlling nutritional status (CONUT) score and the modified Nutrition Risk in Critically Ill (NUTRIC) score.
The modified Glasgow Prognostic Score (mGPS), which combines serum albumin and C-reactive protein (CRP) levels to evaluate the interplay between nutrition and systemic inflammation, has gained attention for its simplicity and prognostic value and hasbeen extensively validated in oncology and other chronic inflammatory conditions, showing robust correlations with survival outcomes. [9,10].
The Prognostic Nutritional Index (PNI), initially developed to evaluate the nutritional status of patients undergoing surgery, combines serum albumin concentration and total lymphocyte count to offer a straightforward measure of nutritional and immune status which can play a role in patient’s outcomes[11-18]. The Controlling Nutritional Status (CONUT) score integrates serum albumin, total lymphocyte count, and total cholesterol
levels, allowing for a nuanced assessment of malnutrition severity [19]. The CONUT score has demonstrated prognostic utility across several clinical scenarios, including cancer [20], coronary artery disease [21], atrial fibrillation [22], acute heart failure [23], and stroke [24]. Recently, Miano et al. highlighted the CONUT score’s high prognostic value for in-hospital mortality, risk of sepsis, and longer hospital stays in Internal Medicine patients, recommending its use as a nutritional screening tool to identify high-risk individuals [25].
The Nutrition Risk in Critically Ill (NUTRIC) score - introduced by Heyland et al. in 2011- integrates age, number of comorbidities, days from hospital to ICU admission, SOFA (Sequential Organ Failure Assessment), APACHE II (Acute Physiology and Chronic Health Evaluation) and interleukin-6 level (IL-6) as an optional variable and is the first validated nutritional score for critically ill patients [26]. To address the practical challenge of measuring IL-6, a simplified version, the modified NUTRIC score (mNUTRIC), was later developed [27]. Subsequent studies have consistently confirmed the NUTRIC and mNUTRIC scores’ role as predictors of adverse outcomes across various clinical settings [28-35]; moreover, all but one research conducted in critically ill cirrhotic patients, in which NUTRIC had a moderate prognostic advantage in comparison to mNUTRIC [36], did not find the NUTRIC Score to be superior to mNUTRIC score in terms of mortality prediction [28-31]
Finally, a novel, non-invasive, and accessible prognostic tool based on the blood urea nitrogen (BUN)-to-serum albumin ratio (BAR) has been proposed. This ratio has shown promise in predicting outcomes in sepsis, as it combines markers of kidney function and protein status to provide a simple yet effective measure of systemic health. Studies suggest that higher BAR values are associated with increased mortality and poorer clinical outcomes, making it a valuable addition to the array of nutritional and prognostic tools in
sepsis management.”
[Please report with numerical data the inputs and outputs of RCTs at lowest risk of bias which tested these scoring systems or compared them. Please also critically review any systematic reviews comparing these indices. You would then be able to specify in detail which ways you planned to improve the measurement of sepsis risks and outcomes].
Comorbidities
Agreed, co-morbidities greatly affect outcomes, especially in seniors.
You reported: “Comorbidities were highly prevalent in this cohort. The most common conditions included hypertension (99/143, 69.2%), diabetes mellitus (62/143, 43.4%), history of coronary cardiovascular events (47/143, 32.9%), and chronic heart failure (56/143, 39.2%). Additionally, 36 patients (25.2%) had chronic obstructive pulmonary disease (COPD), while 81 patients (56.6%) were suffering from chronic kidney disease, including 9 in end-stage renal failure (6.3%). Liver cirrhosis was documented in 21 patients (14.7%). Most patients (122 out of 143 – i.e., 85.3%) presented with more than one associated comorbidity.”
[Please analyse how the co-morbidities and combinations of comorbidities affected outcomes. You reported only vital signs (median systolic blood pressure 110 mmHg, (IQR 40), median diastolic blood pressure 60 (IQR 20), median heart rate 90 bpm (IQR 31), median respiratory rate 20 breaths per minute (IQR 7), and median temperature 37.2°C (IQR 0.8)].
Author Response
Reviewer #1
This is an important study.
Introduction
Please present your literature search so researchers who build on your research can update it.
You described the indices you planned to compare:
“Over the past decades, several scores have been developed to evaluate the nutritional status across different patients’ populations. Among the most clinically relevant and easily applicable in practice are the modified Glasgow Prognostic Score (mGPS), the Prognostic Nutritional Index (PNI), the Controlling nutritional status (CONUT) score and the modified Nutrition Risk in Critically Ill (NUTRIC) score.
The modified Glasgow Prognostic Score (mGPS), which combines serum albumin and C-reactive protein (CRP) levels to evaluate the interplay between nutrition and systemic inflammation, has gained attention for its simplicity and prognostic value and has been extensively validated in oncology and other chronic inflammatory conditions, showing robust correlations with survival outcomes. [9,10].
The Prognostic Nutritional Index (PNI), initially developed to evaluate the nutritional status of patients undergoing surgery, combines serum albumin concentration and total lymphocyte count to offer a straightforward measure of nutritional and immune status which can play a role in patient’s outcomes[11-18]. The Controlling Nutritional Status (CONUT) score integrates serum albumin, total lymphocyte count, and total cholesterol
levels, allowing for a nuanced assessment of malnutrition severity [19]. The CONUT score has demonstrated prognostic utility across several clinical scenarios, including cancer [20], coronary artery disease [21], atrial fibrillation [22], acute heart failure [23], and stroke [24]. Recently, Miano et al. highlighted the CONUT score’s high prognostic value for in-hospital mortality, risk of sepsis, and longer hospital stays in Internal Medicine patients, recommending its use as a nutritional screening tool to identify high-risk individuals [25].
The Nutrition Risk in Critically Ill (NUTRIC) score - introduced by Heyland et al. in 2011- integrates age, number of comorbidities, days from hospital to ICU admission, SOFA (Sequential Organ Failure Assessment), APACHE II (Acute Physiology and Chronic Health Evaluation) and interleukin-6 level (IL-6) as an optional variable and is the first validated nutritional score for critically ill patients [26]. To address the practical challenge of measuring IL-6, a simplified version, the modified NUTRIC score (mNUTRIC), was later developed [27]. Subsequent studies have consistently confirmed the NUTRIC and mNUTRIC scores’ role as predictors of adverse outcomes across various clinical settings [28-35]; moreover, all but one research conducted in critically ill cirrhotic patients, in which NUTRIC had a moderate prognostic advantage in comparison to mNUTRIC [36], did not find the NUTRIC Score to be superior to mNUTRIC score in terms of mortality prediction [28-31]
Finally, a novel, non-invasive, and accessible prognostic tool based on the blood urea nitrogen (BUN)-to-serum albumin ratio (BAR) has been proposed. This ratio has shown promise in predicting outcomes in sepsis, as it combines markers of kidney function and protein status to provide a simple yet effective measure of systemic health. Studies suggest that higher BAR values are associated with increased mortality and poorer clinical outcomes, making it a valuable addition to the array of nutritional and prognostic tools in sepsis management.”
C1. This is an important study.
- Thanks a lot for your positive comments.
Q1. Please report with numerical data the inputs and outputs of RCTs at lowest risk of bias which tested these scoring systems or compared them. Please also critically review any systematic reviews comparing these indices. You would then be able to specify in detail which ways you planned to improve the measurement of sepsis risks and outcomes].
R1. Dear Reviewer thank you for your very positive comments, your deep analysis, and for your advice; in the revised manuscript we tried to introduce and elucidate available previous works comparing the nutritional scores. In particular, the following studies were considered:
- Bodolea, C.; Nemes, A.; Avram, L.; Craciun, R.; Coman, M.; Ene-Cocis, M.; Ciobanu, C.; Crisan, D.Nutritional Risk Assessment Scores Effectively Predict Mortality in Critically Ill Patients with Severe COVID-19. Nutrients 2022, 14, doi:10.3390/nu14102105.
- Zou, S.; Lin, P.; Chen, X.; Xia, L.; Liu, X.; Su, S.; Zhou, Y.; Li, Y. Comparative analysis of six nutritional scores in predicting prognosis of COVID-19 patients. Front Nutr 2024, 11, 1501132, doi:10.3389/fnut.2024.1501132.
- Moghaddam, O.M.; Emam, M.H.; Irandoost, P.; Hejazi, M.; Iraji, Z.; Yazdanpanah, L.; Mirhosseini, S.F.; Mollajan, A.; Lahiji, M.N. Relation between nutritional status a nd clinical outcomes of critically ill patients: emphasizing nutritional screening tools in a prospective cohort investigation. BMC Nutr 2024, 10, 69, doi:10.1186/s40795-024-00869-3.
Of course, any additional suggestion by your side will be welcome.
Comorbidities
Agreed, co-morbidities greatly affect outcomes, especially in seniors.
You reported: “Comorbidities were highly prevalent in this cohort. The most common conditions included hypertension (99/143, 69.2%), diabetes mellitus (62/143, 43.4%), history of coronary cardiovascular events (47/143, 32.9%), and chronic heart failure (56/143, 39.2%). Additionally, 36 patients (25.2%) had chronic obstructive pulmonary disease (COPD), while 81 patients (56.6%) were suffering from chronic kidney disease, including 9 in end-stage renal failure (6.3%). Liver cirrhosis was documented in 21 patients (14.7%). Most patients (122 out of 143 – i.e., 85.3%) presented with more than one associated comorbidity.”
Q2. Please analyse how the co-morbidities and combinations of comorbidities affected outcomes. You reported only vital signs (median systolic blood pressure 110 mmHg, (IQR 40), median diastolic blood pressure 60 (IQR 20), median heart rate 90 bpm (IQR 31), median respiratory rate 20 breaths per minute (IQR 7), and median temperature 37.2°C (IQR 0.8)].
R2. Thank you for your advice. As you can see, we tried to better organize this section of results, also adding new relevant results in the results section and to be commented in discussion section; additional details on comorbidities are provided in section 3.2 (Comorbidities and sepsis-related parameters). In details, in the revised manuscript we have added a logistic regression model drawn to verify the association between each comorbidity and the adverse outcome (exitus) over each other comorbidity; our analysis revealed that Chronic Kidney Disease significantly affect the risk of adverse outcome in septic patients admitted to IM ward (while the others seem not to affect significantly the survival). Moreover, we included the comparisons regarding the nutritional status as evaluated by the scores according to the presence of CKD, highlighting that mNUTRIC (and also BAR) is able to detect the difference in nutritional status better than the other considered scores in this type of patient.
Furthermore, the number of comorbidities affecting the patient was not significantly associated to the outcome (please see the stepwise multivariable regression analysis in the revised manuscript).
I really would thank the Reviewer for his help in improving the manuscript, also on behalf of all coauthors
Reviewer 2 Report
Comments and Suggestions for Authors
This article presents an interesting and valuable exploration of the predictive role of nutritional scores in septic patients. However, it requires significant revisions to enhance its quality and clarity. Below are the key points for improvement:
While the introduction provides a clear background to the problem, it lacks explicitly stated research hypotheses and research questions. These are essential to provide a clear framework for the study. Including such elements would guide readers through the study’s aims and expectations and establish a foundation for the discussion section.
The discussion section should directly address whether the stated research hypotheses and questions have been confirmed or not. This connection is currently missing and would significantly enhance the coherence and impact of the article.
Although the methodology is outlined, it could benefit from stronger statistical support. A more detailed explanation of the statistical analyses performed, including justification for the methods chosen, would increase the robustness of the findings. Additionally, providing clarity on the sample size calculation or power analysis would strengthen the credibility of the study.
The tables, particularly Table 2, are difficult to read and interpret. They should be reformatted to improve readability and ensure that key findings are immediately evident.
The authors reference the Declaration of Helsinki, which is commendable. However, it would be appropriate to clarify whether the study also adheres to the Publication Manual of the American Psychological Association’s guidelines on ethical research practices. This information would reinforce the study’s ethical rigor and transparency.
he conclusions are overly brief and would benefit from expansion. They should provide a comprehensive summary of the findings and implications in simpler language to make the article accessible to a broader audience, including those unfamiliar with scientific terminology.
Author Response
Point-by-point letter addressing the referees’ comments
Reviewer #2
Comment 1. This article presents an interesting and valuable exploration of the predictive role of nutritional scores in septic patients. However, it requires significant revisions to enhance its quality and clarity.
Answer 1. Dear Reviewer thank you for your very positive comments; we tried to address all the issues raised by your side in order to improve our manuscript;
Below are the key points for improvement:
Comment 2. While the introduction provides a clear background to the problem, it lacks explicitly stated research hypotheses and research questions. These are essential to provide a clear framework for the study. Including such elements would guide readers through the study’s aims and expectations and establish a foundation for the discussion section.
Answer 2. Thank you very much for your comment; in the revised manuscript we tried to be clearer about research hypotheses and research questions. In details: “This study seeks to systematically assess the utility of these scores in forecasting key clinical outcomes, including mortality, in septic patients admitted to an internal medicine ward. Moreover, the study further aims to determine specific cutoff values for the scores predicting the clinical outcome - if any, enabling improved predictive accuracy and the establishment of practical thresholds for risk stratification in this patient population. The findings may offer clinicians valuable tools to tailor nutritional interventions and potentially enhance patient outcomes”.
Comment 3. The discussion section should directly address whether the stated research hypotheses and questions have been confirmed or not. This connection is currently missing and would significantly enhance the coherence and impact of the article.
Answer 3. Thank you for your valuable feedback aimed at improving our work. We have revised the text to directly address the research hypotheses and questions, ensuring a clearer connection as per your suggestion in the updated version of the manuscript. In details, we added: “This study evaluated the utility of various nutritional scores in predicting key clinical outcomes, including mortality, in septic patients admitted to an internal medicine ward. Among the variables analyzed, the mNUTRIC score emerged as the strongest predictor of in-hospital mortality, with an AUC of 0.814, underscoring its ability to stratify risk effectively in septic patients. An additional analysis was performed to establish specific cutoff values for the mNUTRIC score in patients admitted to internal medicine wards, aimed at enhancing its predictive accuracy. A threshold of 4.5 points was identified as the optimal cutoff, thereby providing a clear criterion for clinical application”.
Comment 4. Although the methodology is outlined, it could benefit from stronger statistical support. A more detailed explanation of the statistical analyses performed, including justification for the methods chosen, would increase the robustness of the findings. Additionally, providing clarity on the sample size calculation or power analysis would strengthen the credibility of the study.
Answer 4. Thank you very much for your analysis; in the revised manuscript we tried to better explain what we did and why, also by the active support of Prof. Alibrandi (our medical statistics professor). Statistical methods are now described step by step, also including the justification for the methods chosen. About the power analysis: this is a cross-sectional evaluation with a binary outcome; no treatments were compared, nor repeated measures performed; clinical scores were evaluated primarily by univariate analysis, and no sample size calculation is generally needed. About the comorbidity, in order to state if CKD really could be considered associated to the adverse outcome, and evaluating (ex post) the sample size needed by a post-hoc power analysis, the software calculated: <<94 patients are required to have a 80% chance of detecting, as significant at the 5% level, an increase in the primary outcome measure from 32.7% in the control group (death in no CKD pts) to 60.5% in the experimental group (death in CKD patients); however, the OR estimated was 3.139, with an IC: 1.565- 6.295 and a statistical significance as p<0.001. Please, check our changes throughout the text.
Comment 5. The tables, particularly Table 2, are difficult to read and interpret. They should be reformatted to improve readability and ensure that key findings are immediately evident.
Answer 5. Thank you for your valuable feedback regarding the tables, particularly Table 2. We appreciated your input and have taken steps to reformat the table to enhance its readability and ensure that the key findings are presented more clearly.
Comment 6. The authors reference the Declaration of Helsinki, which is commendable. However, it would be appropriate to clarify whether the study also adheres to the Publication Manual of the American Psychological Association’s guidelines on ethical research practices. This information would reinforce the study’s ethical rigor and transparency.
Answer 6. Thank you very much for your input; we checked out for the recommendations of APA, and we added that our study also adheres to the outlined guidelines. This is now clearly stated in the revised manuscript.
Comment 7. The conclusions are overly brief and would benefit from expansion. They should provide a comprehensive summary of the findings and implications in simpler language to make the article accessible to a broader audience, including those unfamiliar with scientific terminology.
Answer 7. Thank you for the advice; in the revised manuscript the paragraph was rebuilt and conclusions were rephrased.
I really would thank the Reviewer for his help in improving the manuscript, also on behalf of all coauthors
Overall, we would to thank the Reviewers and the Editor for their comments and for their valuable inputs; we really appreciate these efforts to improve our research report.
Round 2
Reviewer 1 Report
Comments and Suggestions for Authors
Thanks to the authors for their meticulous attention to the referees' recommendations.
Reviewer 2 Report
Comments and Suggestions for Authors
Thank you for the corrections.